# Rheology as a Tool for Fine-Tuning the Properties of Printable Bioinspired Gels

**DOI:** 10.3390/molecules28062766

**Published:** 2023-03-19

**Authors:** Maria Bercea

**Affiliations:** “Petru Poni” Institute of Macromolecular Chemistry, 41-A Grigore Ghica Voda Alley, 700487 Iasi, Romania; bercea@icmpp.ro

**Keywords:** rheology, bioprinting, 3D construct, viscosity, shear-thinning, yield stress, thixotropy

## Abstract

Over the last decade, efforts have been oriented toward the development of suitable gels for 3D printing, with controlled morphology and shear-thinning behavior in well-defined conditions. As a multidisciplinary approach to the fabrication of complex biomaterials, 3D bioprinting combines cells and biocompatible materials, which are subsequently printed in specific shapes to generate 3D structures for regenerative medicine or tissue engineering. A major interest is devoted to the printing of biomimetic materials with structural fidelity after their fabrication. Among some requirements imposed for bioinks, such as biocompatibility, nontoxicity, and the possibility to be sterilized, the nondamaging processability represents a critical issue for the stability and functioning of the 3D constructs. The major challenges in the field of printable gels are to mimic at different length scales the structures existing in nature and to reproduce the functions of the biological systems. Thus, a careful investigation of the rheological characteristics allows a fine-tuning of the material properties that are manufactured for targeted applications. The fluid-like or solid-like behavior of materials in conditions similar to those encountered in additive manufacturing can be monitored through the viscoelastic parameters determined in different shear conditions. The network strength, shear-thinning, yield point, and thixotropy govern bioprintability. An assessment of these rheological features provides significant insights for the design and characterization of printable gels. This review focuses on the rheological properties of printable bioinspired gels as a survey of cutting-edge research toward developing printed materials for additive manufacturing.

## 1. Introduction

Printing technologies have a high impact on material engineering, which has registered a fast development during the last few years. From simple prototyping for visual inspection purposes to advanced functional fabrication of a large diversity of products, this versatile method offers flexibility, reliability, scalability, customization, durability, and high speed. 3D printing is a fast-emerging technology that allows the manufacturing of three-dimensional solid objects using a specific ink and a digital file and it is largely used in various applications in healthcare, biotechnology, food, agriculture, automotive, locomotive, and aerospace industries, as well as electronics, buildings, consumer goods, etc. [1,2,3,4,5,6,7,8]. During the last few years, hydrogel sheets obtained by 3D printing were submitted to shape transformation with 4D printing technology, controlling the complex dynamics of material in space and time, and inducing stimuli responsiveness [8,9,10,11,12,13]. A high interest is devoted to bioprinting, a technique used for biologically relevant materials which allows accurate positioning of ink and then deposition of cells in a 3D network in order to generate a new tissue that resembles as much as possible the native one. These biomaterials are designed through the synthesis of new compounds using green chemistry (polymers and functionalized peptides) that are then assembled in 3D networks, or by combining various procedures to achieve dynamic and smart network structures (containing natural and/or synthetic molecules) with new functionalities required by the targeted applications [4,5,13,14,15,16,17]. Using specific experimental tools, 3D bioprinting refers to a series of transfer processes used to combine molecules, living cells, scaffolds, and bioactive agents according to a specified configuration to design biological materials (such as organs and living functional tissues for transplantation) [3,10,17,18].

The bioinks are the printing precursors containing living cells and porous, cytocompatible biomaterials with complex flow behavior. Hydrogels as physical, chemical, or interpenetrated networks are largely used for bioinks due to their ability to provide a viable environment for the adhesion, growth, as well as proliferation of living cells. These structures are suitable for the extrusion process presenting shear-thinning and thixotropic behavior. The shear-thinning ability makes the ink printable and allows the shear forces to align the macromolecules along the flow direction, decreasing the viscosity, a measure of their resistance to flow. Thixotropy is a time-dependent behavior (during shear-thinning or viscoelastic tests) that ensures low viscosity and network strength during printing and makes the hydrogels regain their structure and stability after printing. However, the high shear forces or residence times inside the printing needle affect the cell viability, thus among the most important parameters are the shear stress during printing and the cell viability of bioink [19,20,21,22,23,24]. The bioinks pass through a syringe (where the cells are not damaged by flow) to a nozzle with a specific diameter and geometry. The maximum mechanical forces act near the wall of the nozzle where the cell viability decreases exponentially as shear stress rises [25,26]. The main cause of cell death was considered to be the mechanical forces in the syringe that determine the extensional flow at the entrance of the needle [27,28].

A careful evaluation of bioink properties, its flow behavior, and printer characteristics (nozzle diameter, operating temperature, printing pressure, etc.) represent an adequate tool for printing optimization [29]. The applied shear stress must be carefully controlled and maintained at a moderate level (shear stress threshold of 5 kPa [20]) that allows cell differentiation and proliferation. Excessive shear stress disrupts the cell membrane and determines cell death. Also, the wall friction contributed to a decrease in cell viability [20]. The shear dynamics during bioprinting can be evaluated using dimensionless parameters, such as Weissenberg, Oldroyd, Reynolds, or capillary numbers [30].

There were developed various numerical and analytical fluid dynamics models to describe the cells’ effect on material flow, taking into account the influence of the mechanical forces inside of the printing head on the cell viability [20,25,28,29,30,31,32,33,34]. Thus, it was established that the rheology and dynamics of the ink formulations and the applied shear stress represent a critical key factor to judge the material’s printability, printing resolution, and ink integrity [20,35,36,37]. The applied shear stress and shear-thinning behavior under well-established conditions of shear rates must be examined for determining the printability of bioinks. In addition, the viscoelastic parameters (*G*′, *G*″, and tanδ) provide information on the structure and stability of inks before and after the printing process. The printing of highly viscous bioink is difficult to operate [28] and the performances remain modest when high mechanical forces (high pressure) act on integrated cells during bioprinting [24]. The cell viability is influenced by the applied shear stress, the rheological behavior during the bioink flow, and the exposure time. Increasing the shear stress or the residence time inside the needle negatively affects cell viability [19,21].

## 2. Rheological Parameters as Key Characteristics for Extrusion-Based 3D Printing

Rheology is a very useful technique for characterizing inks and evaluating the effects of sterilization or storage conditions on the performance of 3D constructs [23]. Rheological parameters are related to the fundamental processes involved in bioprinting and must be systematically investigated to determine the optimal conditions for biocompatible printing while preserving structural and functional integrity.

Various 3D printing technologies have been developed with different functions, mainly known as extrusion-based printing, material jetting, vat photopolymerization, binder jetting, powder bed fusion, directed energy deposition, and sheet object lamination. The first three processes have been adopted for bioprinting applications using device extrusion of materials, inkjet printing, laser-assisted printing, selective sintering printing, or binder jetting [17,38,39,40]. Recently, other bioprinting techniques with high resolution were developed, such as magnetic printing [41] or electrohydrodynamic jetting [42], however, they are expensive or present low fidelity in 3D constructs.

The required properties of the inks depend on the used printing technique. Each technique presents advantages and limitations [5,10,43]. For extrusion and inkjet-based printing that use nozzles for biomaterial depositing, bioink viscosity plays an important role. Highly viscous materials are not suitable for these technologies [28]. The knowledge of viscoelastic properties is of high interest in bioprinting for determining the adequate mechanical strength to hold the desired structure and the conditions in which solid-like or liquid-like behavior prevails [19,44,45,46].

*Extrusion-based 3D printing* is one of the most versatile and cheap techniques used to print fully functional parts of materials with various colors, such as plastics, food, or living cells [6,47,48]. A variety of gels are used for extrusion-based bioprinting, such as thermosensitive hydrogels (e.g., gelatin, pluronics, methylcellulose), and photocrosslinkable hydrogels [49].

An extrusion bioprinter is mainly composed of a container (usually a syringe [50]) that is loaded with biomaterials, a dispensing head (nozzle) that distributes the biomaterial out, and a material storage mold where the bioink is deposited [51]. Extrudability is correlated with the minimum pressure required for the material to be extruded at a given shear flow rate. This method presents good scalability, and the ability to generate 3D vertical structures, and favors high cell density. The main disadvantages of this printing method are the shear stress effect on cell viability and the reduced printing resolution [14]. A pneumatic force via pressurized air or a mechanical displacement by a screw or a piston allows ink extrusion through the nozzle [26].

There is a high interest in the design of new bioinks, their optimization, and testing in extrusion-based 3D bioprinting. The printing ability and performances of bioinks are evaluated from rheological studies: gelation temperature, shear-thinning behavior, and viscoelastic properties [16]. The viscosity of the extruded inks varies from 10^−3^ Pa s [52] to about 6 × 10^4^ Pa s [14]. The rheological properties are very important parameters for the optimization of extrusion-based printing. The inks should be shear-thinning to be easily extruded through a narrow nozzle, and after printing, the deformation should be minimized to ensure shape fidelity. The thermoreversible gels are suitable materials, rapidly reaching mechanical strength after gelation that ensures a self-healing ability [13,53,54].

The flow behavior and viscoelastic properties of inks are the key characteristics during the extrusion-based process. Some general considerations are applicable to all bioprinting setups, which suppose mainly that the materials are able to flow for shear stress values above a certain limit and recover their structure at the cessation of flow. Thus, information concerning the shear-thinning character, viscosity, yield stress, thixotropy, and viscoelasticity are required for 3D printing inks (Figure 1) [44,55,56,57,58,59,60,61,62].

Rheological properties of inks are strongly dependent on the physicochemical characteristics of inks, namely chemical structure, composition, (supra)molecular architecture, chain length or crosslinking degree, additives, etc. [63,64]. Also, for the gels, the environmental conditions play an important role: pH, ionic strength, temperature, and pressure. All these factors are considered for the material design and allow us to predict its processing behavior. The viscoelasticity of inks depends on the deformation regime during printing and the rheological response of gels is generally time dependent. A deep investigation and an understanding of the rheological properties of each ink are important to enable successful printing [44,65]. For some materials with stimuli-responsive behavior, like thermoreversible hydrogels for food or tissue engineering applications, the gelation/solidify temperature is critical to determine the appropriate printing temperature [13,14,51,66].

Materials with pseudoplastic behavior are suitable for extrusion-based 3D printing. At high shear rates, they easily flow through a moving narrow nozzle. During material deposition, the viscoelastic effects are important and the kinetic energy is converted into elastic and/or dissipated energy. The flow cessation occurs when the shear stress falls below the yield stress, and the edges of the printing constructs are contoured. Thus, to ensure shape fidelity, the thixotropy of the printing materials must be evaluated [57]. Furthermore, bioprinting enables the inclusion of cells or biologically active molecules into the printed constructs in a customized, repeatable, and safe way. Comprehensive rheological characterization and mechanical testing of the bioinks are imperative, to be done in correlation with cell viability assays [19,21,22,25,28,33,34]. Systematic investigations of correlations between these properties and the process parameters allow for predicting biomaterial performances [15,17].

The evaluation of rheological parameters at relevant time scales for inkjet printing is important for establishing the printing parameters of a given material [32,67]. Printable ink has low viscosity during printing and suitable mechanical strength after printing, thixotropic properties, and fast recovery.

The 3D-printing process based on extrusion can be divided into three stages, individualized through the corresponding rheological properties of inks (Figure 1): extrusion (yield stress, viscosity, and shear-thinning behavior), recovery (shear recovery and temperature recovery properties) and a self-supporting stage (viscoelastic moduli and yield stress) [56]. Thus, a series of key rheological parameters are used to characterize the 3D printing process during and after extrusion through the nozzle. However, it is not enough to determine only in part these rheological characteristics. The shear viscosity, shear-thinning behavior, and yield stress suppose destruction of the rest of the structure and these parameters allow an understanding of the ink behavior during printing. The Newtonian viscosity ($\eta_{o}$) and viscoelastic characteristics determined in the linear range of viscoelasticity (*G*′, *G*″, and tanδ) contain information about the material structure, before and after printing. Also, the gelation point (gelation temperature in temperature sweep tests or gelation time at constant temperature) is very clearly determined from dynamic measurements (following the viscoelastic parameters as a function of temperature or time). The thixotropy tests (in continuous or oscillatory shear regimes) allow for the examining the time-dependent recovery of the initial structure and stability after the printing process. The information from the different rheological tests is complementary and must be analyzed as a whole.

A detailed rheological characterization in various shear flow conditions, similar to those existing during printing, allows us to save time, materials, and money. These parameters are discussed further, and their meaning is briefly presented at the beginning of each section to be more easily understood by readers less familiar with the notions of rheology.

### 2.1. Viscosity

Generally, viscosity is the parameter that controls the flow of fluids in well-established conditions of temperature and pressure and it is correlated with the fluids consistency in response to an applied external force [68,69]. The material fluidity is given by the inverse of viscosity. For versatile bioinks, this parameter is tunable by selecting the most appropriate components and composition for a specific application and its knowledge facilitates the usage of the same formulation in different commercial products.

The viscosity expresses the internal friction between the structural entities when they move relative to one another under an applied force, being a measure of material’s resistance to flow or tendency to change its shape. The flow of Newtonian fluids is characterized by a linear relationship between the shear stress (σ [Pa]) and the shear rate ($\dot{\gamma }$ [s^−1^])):(1)$$\sigma =\eta \cdot \dot{\gamma }$$

The ratio of shear stress to shear rate per unit of time is known as the dynamic viscosity, η [Pa·s].

For Newtonian fluids, the viscosity is independent of the shear parameters. However, the materials used in 3D printing have a non-Newtonian behavior when the dynamic viscosity coefficient depends on the shear rate. In such a case, it is called shear viscosity or apparent viscosity. A complete shear flow curve shows three regions (Figure 2):

A Newtonian behavior is observed at low shear rates (σ→0 and $\dot{\gamma }\to 0$). The viscosity corresponding to this region is called zero-shear viscosity, $\eta_{o}$ (also denoted as initial, first Newtonian, or maximum Newtonian viscosity) and its value is correlated with the molecular characteristics, concentration, temperature, pressure, and other environmental parameters (pH, presence of other molecules, etc.). Determined in stationary shear conditions, $\eta_{o}$ contains information about the rest structure of the material. There are materials for which $\eta_{o}$ is not experimentally accessible; these systems flow at very low shear stress values.

A decrease of the apparent viscosity (denoted as $\eta (\dot{\gamma })$) occurs above ${\dot{\gamma }}_{c}$, in the pseudoplastic or non-Newtonian region; ${\dot{\gamma }}_{c}$ depends on the material nature and structure and it is a well-defined value only for low polydispersity; in many cases, this transition from Newtonian to pseudoplastic covers a wide range of $\dot{\gamma }$. The pseudoplastic domain appears for large macromolecules, self-assembling materials, or networks.

For high shear rates, the flow again becomes Newtonian, when the macromolecules are oriented along the flow direction. The apparent viscosity at such high $\dot{\gamma }$ or σ values is known as the second Newtonian viscosity, $\eta_{\infty }$ (also named upper, minimum, or limiting viscosity). For many systems, the $\eta_{\infty }$ value is not determined within the range of experimentally accessible shear rates.

Experimentally, it is not possible to detect all three regions for any material. Newtonian fluids present only the first Newtonian region and the rheological measurements give access to $\eta_{o}$ value. Viscoelastic materials containing macromolecules or supramolecular assemblies in a fluid state present very often the first Newtonian region and the pseudoplastic one. For highly viscous fluids or self-assembling systems, the second Newtonian viscosity ($\eta_{\infty }$) can be experimentally evidenced. For a series of polymeric materials, a complete curve with the three regions is obtained using several rheometers that operate in different domains of shear stress and shear rate. In such cases, the viscosity data may be fitted using the Carreau–Yasuda model:(2)$$\frac{\eta -\eta_{\infty }}{\eta_{o}-\eta_{\infty }}={\left[1+(\lambda \cdot \dot{\gamma }){}^{a}\right]}^{\frac{n-1}{a}}$$
where λ is the relaxation time, *n* is the flow behavior index (the exponent of the power law in the pseudoplastic region), and *a* is Yasuda exponent (for *a* = 2–Equation (2) becomes the Cross model, Equation (3) [70,71]; $\lambda =1/{\dot{\gamma }}_{c}$, ${\dot{\gamma }}_{c}$ being the critical shear rate that delimitates the transition from Newtonian behavior and non-Newtonian flow, i.e., the shear rate at which the viscosity begins to decrease.
(3)$$\eta =\eta_{\infty }+\frac{\eta_{o}-\eta_{\infty }}{1+{\left(K\cdot \dot{\gamma }\right)}^{m}}$$
where *K* has the dimensions of time; *m* is the flow index (dimensionless parameter) and expresses the degree of shear-thinning: a value of *m* closer to zero suggests a tendency to Newtonian behavior and *m* increasing to unity indicates a pronounced shear-thinning character [68].

A more simplified version of the Cross model is the Sisko equation that fits the viscosity at high shear rates, where $\eta_{\infty }$ << $\eta_{o}$ and $K\cdot \dot{\gamma }$ >> 1 [68]:(4)$$\eta =\eta_{\infty }+k\cdot {\dot{\gamma }}^{n-1}$$
where the flow index is given by *n* = 1 − *m* and *k* = $\eta_{o}$/*K* is known as the fluid’s consistency.

The viscosity curve from Figure 2 describes a shear-thinning fluid. A different flow occurs for shear-thickening fluids, when the viscosity increases with increasing the shear rate. This rheological behavior is less frequent and occurs for high concentrations of ceramic suspensions, starch dispersions, foods, or dental composites. For associative polymers, there is a maximum viscosity in moderate shear conditions and the shear-thickening effect disappears at high enough shear rates [72,73,74]. For better control of ink properties, the rheological characterization of precursors should be done [75,76,77].

A higher viscosity and network strength improve the mechanical properties of the ink [78] and the printing resolution [79,80]; in addition, the rheological parameters of inks and printing pressure should be optimized in order to avoid the decrease in cell viability [27,78,81,82]. The behavior of high-viscosity fluids is sometimes associated with that of solids. In the non-Newtonian region, the apparent viscosity can decrease more than 1000 times as compared with the value of $\eta_{o}$. For bioprinting, it is of interest to have a material with low viscosity during printing in order to be easily extruded through a nozzle when a high shear stress is applied. However, the viscosity at the exit from a nozzle, when the flow is stopped, should be high enough to assure the shape fidelity of the construct. Thus, the shear-thinning behavior is analyzed in steady state conditions by measuring the shear viscosity as a function of shear rate. A power-law type dependence (according to the well-known Ostwald-de Waele power-law model) is usually registered for pseudoplastic (non-Newtonian) fluids:(5)$$\eta \;(\dot{\gamma })=k\cdot {\dot{\gamma }}^{n-1}\;\mathrm{or}\;\sigma =k\cdot {\dot{\gamma }}^{n}$$
where *k* (Pa·s*^n^*) is the flow consistency index (the apparent viscosity for a shear rate of 1 s^−1^) and *n* is the flow behavior index and it expresses the extent of shear-thinning during flow.

Usually, the *n* value is around 0.81 for entangled macromolecular systems [83] and it was also reported for physical hydrogels [54]. When increasing shear stresses are applied, high *n* values indicate a pronounced shear-thinning behavior, and the materials can be easily extruded out of the nozzle [13]. High values of *k* and *η* are associated with the hard extrusion of materials from the nozzle; at high shear rates, inhomogeneous flows appear [84] and the nozzle output can be blocked [69,85]. The range of optimum values of viscosity for 3D bioprinters (inkjet, extrusion-based, laser-assisted) is considered between 0.03 Pa·s and over 6 × 10^4^ Pa·s [40].

However, the simple power-law dependence can only be used to describe an ink flow across the shear rate range to which the parameters *k* and *n* were fitted. Equations (2)–(4) are more adequate for shear-dependent fluids. The values of η_o_ and η_∞_ are characteristics of each formulation, being correlated with the physicochemical characteristics of the material.

### 2.2. Yield Stress

Generally, a plastic material presents very small or no deformation when it is subjected to low shear stress. Above a certain level of shear stress, denoted as the yield stress, it starts to flow, a behavior known as visco-plastic [71]. Thus, the yield stress (σ_o_) represents the minimum value of the shear stress (or external force) necessary to be applied in order to break down the structure at rest and initiate the material’s flow (Figure 3). Its value is associated with the mechanical strength of the material [86] and it can be used to evaluate the ability of the bioinks to be extruded [67,87]. Thus, for σ > σ_o_, the material presents fluid-like behavior and, for σ < σ_o_, it preserves the characteristics of a soft solid matter. The better results for yield stress are obtained by presetting the shear force in controlled shear stress tests and they are influenced by the shear history and the used evaluation methods [88].

The experimental data obtained in continuous shear experiments usually fit into yield stress models (Bingham, Hershel–Bulkley, Casson, etc.) [68]. Usually, the Herschel–Bulkley (HB) model is used to describe the flow behavior of inks [32,77]:(6)$$\sigma =\sigma_{o}+k\cdot {\dot{\gamma }}^{n}$$

-For σ < σ_o_, the material exhibits solid-like properties;-For σ > σ_o_ and *n* < 1, the flow behavior is shear-thinning;-For σ > σ_o_ and *n* > 1, the flow behavior is the shear-thickening.

Thus, the low-index *n* illustrates the pattern of the velocity profile for a non-Newtonian flow. With increasing shear rate, the consistency index (*k*) decreases during shear-thinning and increases for shear-thickening behavior. The parameters *k*, *n,* and σ_o_, which describe the shear flow behavior of non-Newtonian fluids depend on their composition and temperature [32].

Some papers reported the existence of two σ_o_ values: a dynamic yield stress corresponding to the destruction of the material structure and a static yield stress that maintains the structure in the disturbed state [89,90,91,92]. Better results for yield stress are obtained by presetting the shear force in controlled shear stress tests. They are influenced by the shear history and the used evaluation methods [88]. Experimentally, the yield stress point of a material can be determined from rheological data using different protocols:-During continuous shear experiments through a double logarithmic plot of shear viscosity as a function of shear stress obtained in shear stress-controlled or shear rate-controlled conditions, the stress ramp tests can be applied for all types of soft gels (for example Figure 4a [93]), but not for hard gels when wall slip appears. Stress can be plotted as a function of viscosity, deformation, or shear rate [2,19,94]. Herschel–Bulkley, Bingham, or Casson fits can be applied to the experimental data to determine σ_o_ [2,69,95];-During oscillatory shear experiments in amplitude sweeps tests, following the viscoelastic moduli as a function of shear stress (as for example Figure 4b). This method is considered very accurate [69,96], however, the interpretation of the experimental data must be carefully done [2,95,97].

Figure 4 shows the rheological data for hybrid hydrogels composed of polymers and proteins [93,98], both methods presented above were used. In continuous shear experiments, yield stress was determined as the shear stress at which the viscosity suddenly changes its dependence on shear stress (Figure 4a). In amplitude sweep tests, the σ_o_ value was obtained from the intersection of the linear plot obtained at low shear stress (in the linear range of viscoelasticity) and those obtained with the experimental data at higher shear stress values (in the nonlinear range of viscoelasticity), as shown in Figure 4b. The last method overestimates the yield stress [2].

**Figure 4 molecules-28-02766-f004:**
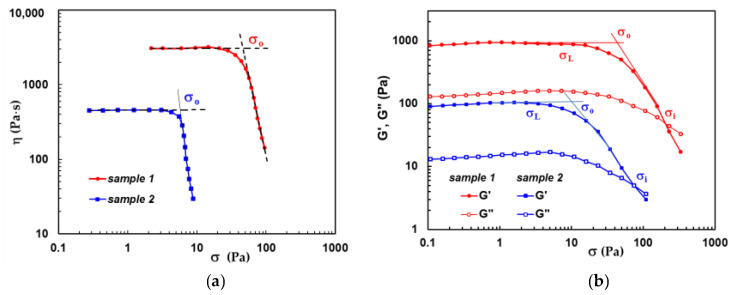
Illustration of yield stress determination from (**a**) the dependence of shear viscosity as a function of shear stress determined in continuous shear conditions [93]; (**b**) from the variation of the viscoelastic moduli as a function of shear stress in amplitude sweep tests. Two hydrogels were considered for this discussion: sample 1 contains 30% BSA and 70% polymer and sample 2 is composed of 70% BSA and 30% polymer. The polymer is a mixture containing 25% pullulan and 75% PVA [98].

In the literature, the evaluation of yield stress in amplitude sweep tests takes into consideration different values from the curves of *G*′ and *G*″ as a function of shear stress or strain [2,95]. Three possible points can be highlighted (Figure 4b):-σ_L_—as the end point of the linear range of viscoelasticity which is characterized by linear dependences of *G*′ and *G*″ on σ;-σ_o_—a cross of the dependences obtained in the linear and nonlinear ranges of viscoelasticity. Above σ_o_, the deformation is very high, and the material is not able to maintain the original structure, starting to show a liquid-like behavior instead of solid-like behavior. This evaluation of σ_o_ seems to be adequate for various materials. Usually, for a very fast evaluation, the shear stress corresponding to a 1% decrease of *G*′ is considered as yield stress value, σ_o_. For very soft samples where the signal is weak, a deviation of 10% of *G*′ is taken into account as a yield point [97];-σ_i_—flow point, as the cross-over point where *G*′ and *G*″ intersect; it is easy to determine experimentally the exact value of σ_i_. Above σ_i_, *G*″ > *G*′, and the material starts to flow.

Even some authors consider either σ_L_, σ_o_, or σ_i_ as yield stress (a more detailed discussion can be found in references [2,95]), all empirical methods take into account that the yield stress is located at the beginning of flow (starting point of non-Newtonian curve) or in the transition zone between linear and nonlinear viscoelastic regime.

A material standing with large deformations before yielding can be considered more stretchable. In this manner, the strain at the yield point may be seen as a measure of the flexibility of materials. When a material is submitted to a stress that is lower than the yield stress, it only undergoes elastic (reversible) strain, without permanent deformation.

Above the threshold value described by σ_o_, the plastic behavior appears, characterized by irreversible deformation of the material. Tomato ketchup is usually considered an example of a material with yield stress; at rest, the structure is as a network without flow; when the bottle with ketchup is shaken very strongly, the structure is destroyed and ketchup flows easily.

### 2.3. Viscoelastic Characterization

Viscosity and yield stress obtained in continuous shear conditions are essential parameters, however, in order to ensure the printability of materials, oscillatory shear experiments are needed to evaluate the viscoelasticity of inks [67,99,100].

The viscoelasticity represents a time-dependent intrinsic characteristic of polymeric materials, bioinks, or extracellular matrix frameworks [19,20,64,101,102]. Viscoelasticity includes two components of deformation: a viscous component, i.e., a gradual deformation induced by the external forces and a continuous reorganization of the fluid molecules, and an elastic component, i.e., after removing the external force the material tends to reestablish the initial structure and shape.

During oscillatory tests, a preset shear strain (γ) and the resulting shear stress (σ) are sinusoidal functions with a time lag, *δ*, between the preset parameter and the registered response. For ideal elastic material, *δ* = 0°, γ, and σ are in phase; for an ideal viscous material, *δ* = 90°; between these two extreme behaviors, there exists a variety of viscoelastic behaviors [71,88].

The oscillatory shear experiments carried out in the linear range of viscoelasticity give access to the viscoelastic moduli, *G*′ and *G*″, which are the real and imaginary components of the complex shear modulus, *G** [83]:(7)$$G*(\omega )=G^{\prime }(\omega )+i\;G^{\prime \prime }(\omega )$$

*G** denotes the material’s overall stiffness under oscillatory shear deformations, as the ratio between the shear stress amplitude, $\sigma_{A}$, and strain amplitude, $\gamma_{A}$:(8)$$G*=\frac{\sigma_{A}}{\gamma_{A}}$$
or
(9)$$G*(\omega )=i\omega \;\int_{0}^{\infty }\;e^{-i\omega \;t}\;G(t)\;dt$$

*G*′ represents the elastic (storage) modulus, being a measure of energy stored (and then returned) during a cycle of oscillation:(10)$$G^{\prime }(\omega )=\omega \int_{0}^{\infty }\sin(\omega t)G(t)dt$$

*G*″ is defined as the viscous (loss) modulus, giving information on energy dissipation during a cycle of oscillation:(11)$$G^{\prime \prime }(\omega )=\omega \int_{0}^{\infty }\cos(\omega t)G(t)dt$$
where *G*(*t*) is the shear relaxation modulus, *ω*—oscillation frequency, and *t*—time.

Thus, the elastic (*G*′) and viscous (*G*″) moduli of bioinks are usually analyzed in the linear range of viscoelasticity (Figure 5a) by following their dependence on oscillation frequency, ω (Figure 5b). The loss tangent (tanδ = *G*″/*G*′) offers information on the viscoelasticity degree of the bioink: tanδ < 1 suggests predominantly elastic (solid-like) behavior and tanδ > 1 is obtained for preponderantly viscous fluids (liquid-like behavior). In practical applications, when tanδ = 100, a fluid can be considered ideally viscous, and for tanδ = 1/100 (0.01) the material presents ideally elastic behavior [88]. For bioprinting, the gel-like structures are of interest and they present *G*′ and *G*″ almost constant, *G*′ > *G*″ (Figure 5b), and tanδ reaches values from 0.01 to 0.1. Within the linear viscoelastic range, the stable network bioinks exhibit constant strength regardless of the oscillation frequency range [69].

High values of σ_o_ and *G*′ ensure the shape fidelity, minimize the ink deformation once it is deposited, and thus avoid the structure collapsing after 3D printing [67,99,100,103,104].

Identification of the linear viscoelastic regime allows to obtain yield strain—the minimum value of deformation before starting the flow as a viscous fluid (*G*″ > *G*′). The yield strain is correlated with the values required to be selected for the force pushing the syringe piston or pressure for pneumatic 3D printing [35].

Uncrosslinked material behaves as a viscous fluid (Figure 5c): at low ω values, the viscous character is predominant (*G*″ > *G*′); for the high-frequency range, the elastic behavior prevails. The oscillation frequency at the crossover point, for which *G*′ = *G*″, allows for determining the longest relaxation time, λ = 1/ω_i_ (expressed in seconds). Usually, for the region of low ω values (ω < ω_i_), the slopes are determined. For Maxwellian behavior, *G*′ scales as ω^2^ and *G*″ as ω^1^. At high frequencies (ω > ω_i_), the elastic modulus becomes independent on ω and its curve shows a plateau value. A decrease in the slope of viscoelastic moduli dependences suggests self-assembling phenomena and small values of these slopes are characteristic of weak gels or low crosslinked networks.

In oscillatory tests, the complex viscosity, $\left|\eta *\right|$, is determined as the ratio between the complex shear modulus: $\left|\eta *\right|=G*/\omega$. For materials with supramolecular structure, such as the gels, the so-called Cox–Merz rule [105] is not valid, i.e., the values of the complex viscosity, η*(ω), obtained in oscillatory shear conditions, are higher as compared with those of shear viscosity, $\eta (\dot{\gamma })$, obtained in the rotational test.

Long-term stability at rest could be evaluated from the analysis of the elastic modulus, *G*′, as a function of time for ω = 0.01 rad/s. Even the curve shows a small variation at the beginning of the test and *G*′ should be constant as a function of time. For a faster assessment, higher ω values can be used between 0.1 rad/s and 1 rad/s [69].

Another possibility to evaluate the viscoelasticity of materials is the analysis of creep and recovery experiments allowing differentiation between solid-like, liquid-like, and viscoelastic behavior (Figure 6).

During creep experiments, a constant shear stress, σ, is applied and it determines a time-dependent deformation which is composed of an instantaneous elastic deformation (γ_E1_), a delayed elastic deformation (γ_E2_) followed by nonlinear deformation, γ(t), that becomes linear when the stationary flow is reached (usually a few tenths of deformation units [54,106]). When the shear stress is removed (at time *t* = $t_{1}$, Figure 6), the recovery starts. Firstly, the instantaneous elastic contribution (γ_E1_) is recovered. Then, time-dependent elastic deformation (γ_E2_) is progressively recovered and it is due to delayed dynamic processes in which the macromolecules are involved. Finally, the viscous contribution is registered (γ_V_) (viscoelastic fluid, Figure 6a). The strain recovery in time depends on the material’s viscosity, chains conformation, connectivity, and ability to return to the equilibrium conformation. Recently, it was shown that the gels with strong intermolecular interactions behave as elastic solids for shear stress values below the yield stress (γ_V_ = 0) and as viscoelastic fluids above σ_o_ (with all three components of the deformation: γ_E1_, γ_E2_, and γ_V_) [54,107]. The energy of deformation is stored during the creep test and when the shear stress is removed, it is completely recovered and used for structure reformation (elastic solid, Figure 6a).

For ideal viscous materials (Figure 6b), during the creep test, there is a continuous increase of deformation in time and, after removing the load, the deformed state is maintained. For such systems, the deformation is permanent and there is no elastic recovery (γ_E1_ = 0 and γ_E2_ = 0), the deformation energy is dissipated as heat and the viscous component dominates the rheological behavior. Such materials are not of interest as inks.

The gels with viscoelastic behavior similar to those presented in Figure 6a are suitable for 3D printing and they must have high strain recovery (*RS*):(12)$$RS(\%)=\frac{\gamma_{e}-\gamma_{t_{1}}}{\gamma_{t_{1}}}\cdot 100$$
where $\gamma_{e}$ is the final strain at equilibrium in the recovery step and $\gamma_{t_{1}}$ represents the strain at time $t_{1}$, when the load is removed and the recovery starts.

The creep (shear) compliance, J(t) [1/Pa], represents the reciprocal value of the shear modulus and it can be calculated as:(13)J(t)=γ(t)/σ

The limiting value of J(t) for t→0 is known as instantaneous shear compliance, $J_{o}$, and allows the determination of the instantaneous shear modulus, $G_{o}$, which is an indicator of the material’s deformability [69]:(14)$$\lim_{t\to 0}J(t)=J_{o}=\gamma_{o}/\sigma =1/G_{o}$$

The instantaneous deformation, γ_o_ [%], is determined from the intersection of the straight line at the beginning of the creep curve with the y-axis.

In addition, creep tests provide access to rheological information in the long-time domain of the relaxation spectra (low-frequency range) that is not accessible using the standard frequency sweep test (0.1 rad/s < ω < 100 rad/s) [108]. Also, these tests can be used for very low shear rates to determine $\eta_{o}$ value, which allows evaluating the average molecular weight of polymers [69]. The slope at the end of the creep domain (steady-state flow), Δγ/Δt, is used to calculate the Newtonian viscosity:(15)$$\eta_{o}=\sigma_{o}\;/\;(\Delta \gamma /\Delta t)$$

The experimental data obtained through the creep and recovery tests can be fitted using Burgers model with four parameters (consisting of two elements, Maxwell and Kelvin–Voigt, combined in series) [69,88,109,110,111]:(16)$$J(t)=J_{o}+J_{1}\left[1-\exp(-t/\lambda {}_{r})\right]+t/\eta_{o},\;\mathrm{for}\;t\;\leq \;t_{1}$$
(17)$$J(t)=J_{1}\left[\exp((t_{1}-t)/\lambda {}_{r}))-\exp(-t/\lambda {}_{r})\right]+t_{1}/\eta_{o},\;\mathrm{for}\;t\;>\;t_{1}$$
where $J_{1}$ [1/Pa] is the compliance associated with the Kelvin–Voigt element, $\eta_{o}$ [Pa·s] is the viscosity of the Maxwell dashpot, $\lambda {}_{r}$ [s] is the retardation time associated with the Kelvin–Voigt element and $t_{1}$ is the time when the stress was removed.

Burgers model uses a relatively large number of parameters (four parameters in Equations (16) and (17)), and it was not always able to describe the two steps of creep and recovery for inks [111,112].

Another approach is fractional calculus [113], which is able to predict the complete viscoelastic behavior. The application of the fractional calculus allowed a reasonable fitting of both the creep and recovery steps [111,112]:(18)$$J(t)=\frac{\gamma (t)}{\sigma }=\frac{1}{\Gamma (\alpha +1)}\left[\lambda_{1}\;t^{\alpha }\;H(t)-\lambda_{2}{\left(t-t_{1}\right)}^{\alpha }\;H(t-t_{1})\right]$$

The parameter, α, provides an indication of the degree of elasticity and varies between 0 and 1: α = 0 for ideal elastic materials and α = 1 for ideal viscous behavior; $\lambda_{1}$ and $\lambda_{2}$ are the inverse of the elastic modulus during creep and recovery steps, respectively; Γ is the gamma function and *H*(*t*) is the step function defined as:(19)$$H(t)=\left\{\begin{array}{l} 0,\;\;if\;t\leq t_{1}\\ 1,\;\;if\;t>t_{1} \end{array}\right.$$

Equation (18) can be written as:(20)$$J(t)=\frac{\gamma (t)}{\sigma }=\left\{\begin{array}{l} \frac{1}{\Gamma (\alpha +1)}\lambda_{1}\;t^{\alpha },\;\;t\leq t_{1}\\ \frac{1}{\Gamma (\alpha +1)}\left[\lambda_{1}\;t^{\alpha }-\lambda {}_{2}(t-t_{1})\alpha \right],\;\;t>t_{1} \end{array}\right.$$

For materials that exhibit high deformation during the creep test, the applied shear stress influences the structure at rest, and they may have a high resistance during recovery, thus $\lambda_{1}>\lambda_{2}$. The analysis of the ($\lambda_{1}-\lambda_{2}$) difference provides information on recovery behavior: a large difference suggests a high degree of permanent deformation and significant changes in the internal structure of the material; a small ($\lambda_{1}-\lambda_{2}$) difference is desired for printable gels indicating that they present a high ability to recover their structure after deformation [111,112]. The recovery time, *t_recovery_*, is also analyzed and it should not exceed several tens of minutes.

### 2.4. Recovery Time and Self-Healing Ability

An ideal printable material presents high thixotropy or self-healing ability, the viscosity (or other parameters) becomes low quickly when high shear forces are applied and it recovers very quickly the initial value after removing the load. The self-healing ability is given by the presence of dynamic reversible bonds in a polymer network. They are destroyed when high forces are applied and then reestablished in time. From a structural point of view, it is important that the intermolecular interactions that ensure the junction points recover rapidly and to a high extent after printing. The self-healing ability is given by the presence of dynamic reversible bonds in a network which are destroyed when high forces are applied and then reestablished in time. Such reversible bonds could be noncovalent physical bonds (such as hydrogen bonding, hydrophobic, host–guest, electrostatic interactions, π–π stacking, and metal coordination) [114,115] or covalent bonds (such as disulfide or imine bonds, boronic esters, reversible diarylbisbenzo-furanone crosslinking, and urea–urethane exchange) [115,116,117,118,119].

Thixotropic properties and recovery times can be investigated during the steady shear or oscillatory regime of deformation [88]. The stable microstructure of a material at rest is perturbed by shear forces. Then, when the shear forces are ceased, the material structure tends to recover the equilibrium state. The ability to restore the original structure is known as self-healing and the time required for reaching again the equilibrium state or the initial structure from the rest state is considered the recovery or self-healing time [2,77,120]. The material recovery in time is known as thixotropy [121,122,123] and it is evidenced by the reversible time-dependent structural changes after a deformation cycle which includes low—high—low levels of deformation. These changes are reflected by the evolution in time of the shear viscosity ($\eta (\dot{\gamma })$) or viscoelastic parameters (*G*′, *G*″, and tanδ) [123,124,125]. A sample that was in equilibrium at rest is submitted to shear flow from low to high shear rates, then the shear rate is decreased and the recovery of viscosity in time is followed [123]. Similarly, the viscoelastic parameters are monitored in a third test interval thixotropy (Figure 7) [125]. However, the recovery time should be optimized since it influences cell incorporation. Instantaneous recovery induces a heterogeneous cell distribution, very long recovery time determines cell sedimentation, heterogeneous distribution, or poor shape retention [126]. The optimal recovery time should be neither too long nor too short.

Tailorable properties can be induced by multiple crosslinking strategies using combined methods [2,13,127,128,129,130,131,132,133], the incorporation of dynamic reversible bonds [134,135], or the design of hybrid polymer–peptide materials [54,93] that allow different relaxation mechanisms during bioprinting and ensure the shape fidelity. Recommended recovery times for bioprinting are on the order of tens of seconds (Figure 6a and Figure 7), for example above 85 % recovery of elastic modulus or viscosity within up to 30 s [77,125,136,137]. After extrusion and recovery, the bioink must resist external forces such as the weight of stacking layers that could determine the deformation and poor shape fidelity. The elastic behavior of materials determines the structure recovery after the load was released. During deformation, the material is able to store energy by expanding or stretching the structural entities involved in the supramolecular assembly, without destroying the structural integrity of the material. The stored energy is the driving force to restore the initial structure and to assure the shape fidelity of the material after the applied load was released [88].

The thixotropy test can be corroborated with other rheological behaviors, such as full elastic recovery typical to solids (Figure 6a) in creep and recovery tests or viscosity loops at increasing and decreasing shear rates in continuous shear experiments [54]. The self-healing concept is now frequently used to characterize bioinks and it refers to the ability of materials to regenerate their structure after being submitted to deformation. For example, Figure 8 illustrates the porous/macroscopic structure and rheological behavior of a self-healing 3D printing ink with dynamic covalent bonds (boronic ester), composed of 5% (wt.) poly(N,N-dimethylacrylamide-3-acrylamidophenilboronic), 2.5% (wt.) poly(glycerol monomethacrylate), and 0.5% (wt.) poly(dopamine) coated chemically reduced graphene oxide [115]. Two pieces of this sample (marked as **a** and **b** in Figure 8) could heal in about 20 s, and the recovered gel showed good resistance to external forces.

## 3. Rheology as a Prerequisite for Bioink Formulation and Optimization

The selection of materials for the ink formulation is based on their intrinsic properties that influence printability. Tailored bioinks were obtained using composite biomaterials with synergetic properties and improved printability. When multicomponent systems are used to design complex inks, physical interactions, self-assembling phenomena, temperature, or shear-induced phase transitions, or even possible chemical reactions, must be taken into account. For each ink formulation, a preliminary rheological investigation of the materials is a necessary step [77] since the appropriate viscoelastic properties are crucial for optimum printability. There are many studies evaluating the performances of different bioinks—a mixture of cells and biomaterials—and many attempts to improve printability. Various materials were investigated as bioinks and they are presented in comprehensive reviews (see for example [3,4,5,6,7,8,9,11,12,14,138]). Suitable 3D inks were obtained using polysaccharides and proteins [139,140,141,142], methacrylic polymers, vinyl derivatives, and glycols [44,143,144,145], and ceramics or cement [6,39,95,146,147,148]. In many cases, the inks are multicomponent systems (smart polymers in solution or gel state), and the optimization of printable formulations is very often done by rheology [35,36,47,56,149,150,151,152,153,154,155].

Polysaccharides are largely used for bioprinting [4,12,139]. Among the most commonly used polysaccharides for bioinks design are found chitosan [156,157,158], alginate [159,160], hyaluronic acid [133,161,162,163,164,165], pullulan [166], agarose [167,168], and carrageenan–xanthan–starch [56]. Furthermore, proteins [37] and peptides [169] are incorporated into bioinks. The most frequently used are collagen [170,171,172], gelatin and its derivatives [87,154,173,174,175], fibrinogen in soft or hard tissues [176,177,178], egg white [179], etc. In addition, due to superior mechanical properties, synthetic polymers are potential candidates for bioprinting, such as polycaprolactone [180], poly(ethylene glycol) [144,145], and polyurethanes [54,181,182]. Additives are also used for changing the hydrogels’ rheology, for example nanofibrillated cellulose is added to alginate gels in order to improve the shape fidelity of the bioink [183,184,185,186].

Rheological behavior, with respect to physicochemical composition, temperature, and aging, gives the possibility of designing versatile viscoelastic ink by determining the critical parameters of the inks and avoiding undesirable effects. The polymer structure (molecular weight, concentration, supramacromolecular architecture, etc.) presents a high importance in conferring the required rheological parameters of inks.

The rheology of polymer-based dopant-source inks is decisive for obtaining defect-free droplets [187,188]. Determination of the shear viscosity is not enough for the jetting performances, short timescale parameters are also necessary to evaluate the stability of ink droplets. Thus, the jetting behavior was correlated with the elastic modulus determined at 5 kHz (using a special device, a piezoelectric axial vibrator rheometer) [188]. The presence of air bubbles is detrimental to inkjet printing as they can prevent ink ejection from the nozzles [187].

Gelatin methacrylate (GelMA)-based bioinks are frequently used for extrusion bioprinting, presenting suitable rheological and photo-crosslinking characteristics [44,189,190,191,192]. Low concentrations (<5% *w*/*v*) of GelMA are favorable for cells, but such systems are not printable. In printable constructs, usually GelMA is mechanically stabilized through a combination with polysaccharides [192,193], proteins [194], or synthetic polymers [44,49,78]. For precise and controlled deposition, low viscosity and thus low-resistance to flow bioinks are preferred, being able to promote the migration and growth of the encapsulated cells. Viable and functional bioinks were obtained from GelMA and alginate [193] or gelatin [194] (Figure 9).

Nanocellulose-based materials are very attractive for 3D bioprinting due to the unique characteristics of cellulose, namely printability, mechanical strength, biocompatibility, biodegradability, and high cell viability [195,196,197,198]. At high shear rates, nanocellulose chains in suspension align and form shear-thinning or shear-thickening thixotropic systems with rapid recovery [199,200,201].

The addition of nanocellulose into the bioink formulations improved considerably the printability [35,196,198]. Extrusion of tripeptide–alginate–cellulose bioinks resulted in stable 3D structures for cellulose concentrations greater than 10%, and only above 40% cellulose content the inks were able to stack multiple printed layers, without fusing [35] (Figure 10). The bioinks’ optimization was done on the basis of rheological measurements. The inks presented shear-thinning behavior with viscosity around 10^3^ Pa·s at a shear rate of 1 s^−1^, and the printed scaffolds recovered their structural integrity after the printing process. The yield strain was 45%, whereas the elastic modulus (*G*′) reached values between 2.2 × 10^4^ and 2.4 × 10^4^ Pa, and tanδ = 0.3 (in the linear range of viscoelasticity).

The printability of a variety of multicomponent food inks was extensively investigated through different rheological tests [36,56,60,67,202,203,204,205]. It was shown that the addition of Konjac improves the mechanical and rheological behavior of agar gels and makes the extrusion smoother [202]. For the food industry, research was oriented to improving the quality of life through efficient use of the existing food materials but also to design healthy and customized foods with various sensory properties for personalized nutrition [67,206]. By adding xanthan gum into κ-carrageenan ink, the printability and shape retention performances were improved; the gelation temperature and time, viscosity, yield stress, and viscoelastic moduli increased, whereas the shear-thinning behavior was enhanced [56].

Haider et al. [207] reported recently a smart thermosensitive 3D network formed by a diblock copolymer of approximately 100 repeating units containing hydrophobic poly(2-N-propyl-2-oxazine) (pPrOzi) and hydrophilic poly(2-ethyl-2-oxazoline) (pEtOx), suitable for extrusion-based 3D printing (Figure 11). The rheological investigations evidenced the gelation as a function of temperature and concentration, the viscoelastic and shear-thinning behavior, as well as the ability of rapid structure recovery, the hydrogel being suitable for tissue engineering applications (cell viability ≈ 97%).

3D printing is considered a versatile approach, suitable for achieving pharmaceuticals from polymeric matrices of well-defined porosity, geometry, and size for controlled-release dosage forms [208,209,210].

According to the high number of published papers (for example: [1,2,3,4,5,6,7,8,14,39,41,44,48,67,78,131,157,159,163,164,165,166,167,170,176,179,182,183,193,207,208,209,210,211,212]), there is a huge interest for printing techniques to design products of various shapes and functionalities for a wide area of applications, from regenerative medicine and tissue engineering, foods, pharmaceuticals, biotechnology, biomedical sensing and imaging, body robotics, electronics, and biosensors to civil constructions, aeronautics, and space missions. Practically, 3D printing, known also as additive manufacturing, can be regarded as a revolutionary and economic strategy to produce materials.

3D bioprinting is a versatile approach that allows for the creation of tailored architectures capable of mimicking the extracellular matrix. The biological materials can be deposited layer-by-layer in adequate conditions to generate customized tissue scaffolds or organs. Thus, 3D bioconstructs provide the matrix able to promote cell attachment, proliferation, and differentiation for the regeneration or replacement of functional tissues or organs [213].

Systematic rheological investigation of bioinks and analysis of the viscoelastic parameters is mandatory for the optimization of 3D bioprinting in order to ensure high quality, shape fidelity, and cell viability of the 3D constructs.

## 4. Concluding Remarks

Due to their network structure and properties similar to natural tissues, various gels are suitable biomaterials for 3D bioprinting, providing a viable environment for cell adhesion, growth, and proliferation. From a rheological point of view, the most appropriate bioinks behave as solid-like fluids, with yield stress, shear-thinning, and self-healing ability (Figure 1). Controlling the viscoelastic features of the biomaterials represents the key factor in bioprinting [16,19,21,28,29,30,33,56,62,66,87,99,147,171].

Each printing technique requires specific rheological properties which allow for the obtaining of performances in optimum conditions. As for example, a highly viscous material is difficult to be processed. Due to the high stiffness, it requires high forces and thus it is not an optimum environment for cell culture. A weakly structured ink flows easily, but it cannot ensure the material’s shape fidelity or the required mechanical properties. The flow and viscoelastic characteristics govern the printability and these considerations are taken into account for the a bioink design [19,35,37,55,56,67,77,115,188,194,214,215]. The rheological tests presented in this paper provide crucial information concerning the printability of the materials during different stages of printing, according to Figure 1. In addition, the viscoelastic parameters are very useful for assessing the printability of stimuli-responsive materials (pH, temperature, electrical or magnetic field, light, biomolecules, mechanical forces, etc.) [9,67,216]. As for example, the temperatures of interest are correlated with the conditions of storage, use, or those developed during bioprinting. All these rheological features need to be analyzed for bioink optimization. Tables 1 and 2 of [67] or Tables I and II of [217] offers examples of such analysis for bioinks used for 3D printing (for applications in the food industry [67] or materials for tissue engineering and regenerative medicine [217]).

Mechanical characterization is necessary to assess the shape stability of the printed structures in the conditions of their use. The required mechanical load is correlated with the rheological characteristics of the ink [214] and with the nozzle geometry [218]. Mainly, the inks’ functionality depends on their printability [16,19,21,28,30,33,37,57,66,87,99,147,172,181,186,188,203,219,220] and shape fidelity [53,54,67,183,184,185,186,221,222]. For bioinks, the assessment of biocompatibility is also required [21,22,195,196,223]. They should be biodegradable [182,216,224], able to provide 3D tissue-engineered constructs with the appropriate biological environments and biomechanical functions of the native tissue within the body (bone, skin, osteochondral or cardiac reconstruction, etc.) [224]; also, they create space, being the replaced by new tissue created in time [17,213,223]. Every ink has specific requirements for its viscosity which depend on the printing process. The printability and shape fidelity of a printed scaffold is improved by controlling the inks’ viscosity, which should be carefully optimized for each application [15,215,225]. As an example, if the applied extrusion pressure is too high for a viscous bioink, this will damage the living cells. Quantitative rheological measurements on the ink are required for correct printability assessments under various conditions similar to those existing in the targeted use. The shear-thinning behavior is a must for the bioinks used in extrusion-based printing to allow the flow through the needle [15,120,226]. Furthermore, upon material deposition on a substrate, the plotted shape must be preserved. Thus, it is very important to ensure the recovery of the ink structure after the flow cessation and long-term shape fidelity. The ink’s biocompatibility will enable a long-term culture of the encapsulated cells. The cell viability tests should be performed for a minimum of two weeks following the proliferation and protein synthesis assays, and it should be checked if the printing process affects the cellular function or phenotype [23].

The final characteristics of the printed material may correspond with the functions for which the construct was designed [23,78,227]. Very often, the inks fulfill only in part these demands since it is difficult to incorporate these characteristics all together in a perfect bioink structure. In some cases, a sacrificial mold (for example pluronics or gelatin) is used to create two hydrogels’ 3D architecture (the second network may be, for example, UV crosslinked methacrylate) able to fill any arbitrary geometry. Then one component is removed (pluronic is washed below its gelation temperature) and a stable matrix with controllable pore size, that favors cell attachment, is obtained [15,228,229]. Standards are also required for sustainable advancement in bioprinting. They need to cover all activities, from laboratory procedures to the implementation of biological processes [23].

The modeling of rheological behavior is necessary to develop general models for all materials or categories of materials used for bioprinting. Different approaches are currently used, from empirical models to phenomenological ones [32,34,84,111,112,113,122,217,230].

In conclusion, due to the complexity and diversity of printable materials, it is difficult to incorporate in one single rheological approach a complete viscoelastic characterization of bioinks, to define absolute criteria or clear limits of rheological parameters. The experimental measurements in various flow conditions are necessary for the optimization of the materials and for selecting the parameters that influence the printability of each system.

## Data Availability

Data are available on request.

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
