# Peer review of "Rheology as a Tool for Fine-Tuning the Properties of Printable Bioinspired Gels"

_molecules, 2023, doi:10.3390/molecules28062766_

Round 1

Reviewer 1 Report

The review provides a good overview over most of the relevant rheological properties that are necessary for biofabrication. It extends the scope to more than extrusion-based systems and also discusses other techniques like inkjet deposition. For either set of techniques, rheology is extremely important, but for each a different subset of rheological properties as more significance.

The review starts by giving a basic introduction into additive manufacturing techniques and then builds a description of the rheological characteristics on this basis. It’s a very good overview over many important design considerations for liquid formulations. Many of the parameters are of direct consequence and very important to monitor and tailor for successful 3D printing. This is mentioned all throughout the manuscript and is certainly the motivation for a research paper of this scope.

There are some things that should be improved however, before it could appeal to a broad audience. The critical points that should be addressed can be divided into three categories, structural points, content points and language points.

I will start with the most important, which is the structural one:

1.         The order of the individual chapters could be improved by something like a “definitions” chapter prior to the main rheological part. The main rheological parameters like “shear-thinning”, “yield stress” and “visco-elasticity” should be explained in a quick overview with simple terms. Then especially the inexperienced reader is prepared for these terms.

2.         In the end there is Scheme 1 which feels like it should be presented right in the beginning of the review. This can be used together with point #1 to present the reader with a clearer overview of the relevant topics of this review.

3.         In the Figure 10, the rheological parameters relevant for extrusion systems are directly shown. The Figure is excellent to show precisely where in the whole system which rheological parameter is important. This also should be presented much earlier, after the explanation of the rheological parameters (point 1#) and the overview of Scheme 1 (point #2). Then at least for such extrusion-based systems the following chapters would be much clearer.

4.         A serious consideration should be made, if the review really should be so broad in terms of the additive manufacturing techniques. Most of the points discussed have their relevance for extrusion-based systems and they do not hold for inkjet systems. Almost the only connection point in this review is the necessity to tailor their rheological properties. But what that means respective to the technique is never discussed explicitly, it is just shown exemplary for extrusion-based systems. That may give the false impression that the direction in which to tailor the liquid formulation is identical, no matter what the system in question is.

5.         Alongside point #4 it is questionable if the introduction to additive manufacturing in Chapter 2 should be extended to so many different techniques. With the rest of the review, it is not clear why techniques like Powder Bed Fusion, Directed Energy Deposition, Vat Polymerization and Binder Jetting should be discussed. None of the rheological parameters discussed later have a large impact on these techniques or their impact is so different from what is discussed in the later chapters that they become detached content-wise.

6.         Another chapter should be introduced, that details to what extent the individual rheological properties influence each other. At some points it is written that they cannot be tailored independently, yet at many other places the point is made that the inks should be tailored rheologically. If the rheological parameters depend on each other, at least a small subchapter should be presented that details to what extent this is the case. Otherwise, the reader has no chance in tailoring their gels for rheology.

The next part of the review is related to the content, which can be improved in some places.

7.         The Figures 6 and 7 should be combined to a single Figure. Additionally, in this chapter, it should be made clearer why and how stress relaxation does influence the printing quality. It is not very clear to the reader what happens to a gel after being printed and why this recovery should be considered. For instance, after reading I am not certain if I want structural recovery in my liquid formulations or not. Additionally, it should become clearer how structural recovery can be achieved in gels. The gel represented in the Figures 6 and 7 is just labeled as a “hydrogel” suggesting that any gel will show the structural recovery which renders considering the gel recovery as an important rheological parameter irrelevant.

8.         In Figure 10, the right side is irrelevant for the comprehension of the paper and should be cut in favor of the left part which is very important and should be presented earlier (cf. point #3).

9.         The size of Figure 9 should be improved. The shown details (scale bars, numbers etc.) are too small to read.

10.     In the chapter 2.3 for Vat Polymerization the technique is described as being used with “laser, kight or ultraviolet (UV).”. This suggests that there are different sources for the polymerization, when it is all electro-magnetic radiation, just the wavelength and the light sources are different. The question remains, if this chapter will stay (cf. point #5) but this should be changed to make it clearer how the polymerization is triggered.

11.     In chapter 2.5 Powder bed fusion a “head thermal print” is described. What is this or is “thermal printhead” meant instead. Then again, the question arises what that is. But like for point #10 it is questionable if the chapter should remain at all.

Author Response

The review provides a good overview over most of the relevant rheological properties that are necessary for biofabrication. It extends the scope to more than extrusion-based systems and also discusses other techniques like inkjet deposition. For either set of techniques, rheology is extremely important, but for each a different subset of rheological properties as more significance.

The review starts by giving a basic introduction into additive manufacturing techniques and then builds a description of the rheological characteristics on this basis. It’s a very good overview over many important design considerations for liquid formulations. Many of the parameters are of direct consequence and very important to monitor and tailor for successful 3D printing. This is mentioned all throughout the manuscript and is certainly the motivation for a research paper of this scope.

There are some things that should be improved however, before it could appeal to a broad audience. The critical points that should be addressed can be divided into three categories, structural points, content points and language points.

I thank the reviewer for the careful analysis of the manuscript and all the critical points that helped me to considerably improve the content of the work. In the revised manuscript I took into account all these constructive comments.

I will start with the most important, which is the structural one:

  1. The order of the individual chapters could be improved by something like a “definitions” chapter prior to the main rheological part. The main rheological parameters like “shear-thinning”, “yield stress”, “thixotropy” and “visco-elasticity” should be explained in a quick overview with simple terms. Then especially the inexperienced reader is prepared for these terms.

The introduction of a new subchapter would have made it even more difficult to understand the text for those who are not familiar with rheology. I considered that it is more practical to explain each parameter as it was discussed and to include some references where this information is presented in detail.

  1. In the end there is Scheme 1 which feels like it should be presented right in the beginning of the review. This can be used together with point #1 to present the reader with a clearer overview of the relevant topics of this review.

Thanks to the referent for the suggestion, the scheme was included at the beginning of review.

  1. In the Figure 10, the rheological parameters relevant for extrusion systems are directly shown. The Figure is excellent to show precisely where in the whole system which rheological parameter is important. This also should be presented much earlier, after the explanation of the rheological parameters (point 1#) and the overview of Scheme 1 (point #2). Then at least for such extrusion-based systems the following chapters would be much clearer.

It is a very good suggestion; it was taken into account in the revised manuscript.

  1. A serious consideration should be made, if the review really should be so broad in terms of the additive manufacturing techniques. Most of the points discussed have their relevance for extrusion-based systems and they do not hold for inkjet systems. Almost the only connection point in this review is the necessity to tailor their rheological properties. But what that means respective to the technique is never discussed explicitly, it is just shown exemplary for extrusion-based systems. That may give the false impression that the direction in which to tailor the liquid formulation is identical, no matter what the system in question is.

This part was revised.

  1. Alongside point #4 it is questionable if the introduction to additive manufacturing in Chapter 2 should be extended to so many different techniques. With the rest of the review, it is not clear why techniques like Powder Bed Fusion, Directed Energy Deposition, Vat Polymerization and Binder Jetting should be discussed. None of the rheological parameters discussed later have a large impact on these techniques or their impact is so different from what is discussed in the later chapters that they become detached content-wise.

This part was revised. Indeed, for some printing techniques, the impact of rheological properties is not so evident. All the techniques were initially discussed for an overview, but their relevance in the context of this work does not justify their presentation.

  1. Another chapter should be introduced, that details to what extent the individual rheological properties influence each other. At some points it is written that they cannot be tailored independently, yet at many other places the point is made that the inks should be tailored rheologically. If the rheological parameters depend on each other, at least a small subchapter should be presented that details to what extent this is the case. Otherwise, the reader has no chance in tailoring their gels for rheology.

This comment is very constructive and indeed the initial text leads to confusion.

The main idea was reformulated in the following part:

<< However, it is not enough to determine only in part these rheological characteristics. The shear viscosity, shear-thinning behavior and yield stress suppose a destruction of the rest structure and they allow understanding the ink behavior during printing. The Newtonian viscosity (ho) and viscoelastic characteristics determined in the linear range of viscoelasticity (G’, G” and tand) contain information about the material structure, before and after printing. Also, the gelation point (gelation temperature in temperature sweep tests or gelation time at constant temperature) is very clearly determined from dynamic measurements (following the viscoelastic parameters as a function of temperature or time). The thixotropy tests (in continuous or oscillatory shear regime) allow examining the time-dependent recovery of the initial structure and stability after printing process. The information from the different rheological tests is complementary and must be analyzed as a whole. >>

The next part of the review is related to the content, which can be improved in some places.

  1. The Figures 6 and 7 should be combined to a single Figure. Additionally, in this chapter, it should be made clearer why and how stress relaxation does influence the printing quality. It is not very clear to the reader what happens to a gel after being printed and why this recovery should be considered. For instance, after reading I am not certain if I want structural recovery in my liquid formulations or not. Additionally, it should become clearer how structural recovery can be achieved in gels. The gel represented in the Figures 6 and 7 is just labeled as a “hydrogel” suggesting that any gel will show the structural recovery which renders considering the gel recovery as an important rheological parameter irrelevant.

Figures 6 and 7 were combined in a single figure (new Figure 7).

The creep and recovery tests are useful to illustrate the time dependent viscoelastic response of a material submitted to a shear stress. The author considers these tests relevant for printing process, even they are less often discussed for inks.

The discussion concerning the creep and recovery behavior was revised and improved.

  1. In Figure 10, the right side is irrelevant for the comprehension of the paper and should be cut in favor of the left part which is very important and should be presented earlier (cf. point #3).

Figure 10 - new Figure 1 in the revised manuscript – contains also the right side for visual impact - the reader clearly sees that a printing process is being discussed.

  1. The size of Figure 9 should be improved. The shown details (scale bars, numbers etc.) are too small to read.

The size of Figure 9 has been increased and the clarity has been improved.

  1. In the chapter 2.3 for Vat Polymerization the technique is described as being used with “laser, kight or ultraviolet (UV).”. This suggests that there are different sources for the polymerization, when it is all electro-magnetic radiation, just the wavelength and the light sources are different. The question remains, if this chapter will stay (cf. point #5) but this should be changed to make it clearer how the polymerization is triggered.

According to the previous comments, this part was removed from the revised manuscript.

  1. In chapter 2.5 Powder bed fusion a “head thermal print” is described. What is this or is “thermal printhead” meant instead. Then again, the question arises what that is. But like for point #10 it is questionable if the chapter should remain at all.

As mentioned above, this part was removed from the revised manuscript.

  1. In chapter 3.3, the relation to 3D bioprinting is not clear. The relationship between the time dependent shear (also creep and recovery) should be made clearer to the reader. Why is it important for bio printing researchers to understand these physical relationships? How can creep influence bioprinting and how does recovery play into a finished construct. The formulas will be unnecessary for a casual reader who just wants to understand the relationship between rheology and printability.

The discussion concerning the creep and recovery behavior was revised and improved.

  1. In Figure 4c) the Y-Axis reads G’, G’’ and tan delta, but the plot only shows G’ and G’’. Additionally, it is not clear, why G’ depends quadratically and why G’’ depends linearly on omega.

Here it was an error; I thank the reviewer for the correct comment. Figure 4 (new Figure 5c) was revised.

<< Uncrosslinked material behaves as a viscous fluid (Figure 5c): at low w values the viscous character is predominant (G” > G’); for high frequency range, the elastic behavior prevails. The oscillation frequency at the crossover point for which G’ = G” allows to determine the longest relaxation time, l = 1/wi (expressed in seconds). Usually, for the region of low w values, the slopes are determined. For Maxwellian behavior, G’ scales as w2 and G” as w1. At high frequences, the elastic modulus becomes independent on w and its curve shows a plateau value. A decrease of the slope suggests self-assembling phenomena and small values of these slopes are characteristic to weak gels or low crosslinked networks.

In oscillatory tests, the complex viscosity is also determined as:

                                                                                                                                                                    (11)

For materials with supramolecular structure and gels, the so-called Cox-Merz rule [105] is not valid, i.e., the values of the complex viscosity, h*(w), obtained in oscillatory shear conditions, are higher as compared with those of shear viscosity, , obtained in rotational test.  >>

  1. In chapter 3.4, the word “crosslinks” is used in a context where rather intermolecular forces or bonds should be used. Crosslinks are usually covalent or ionic (so very strong) bonds, but in this context just attractive bonds like hydrogen- or Van-der-Waals bonds are meant.

I thank the reviewer for this comment. The word <<crosslinks>> was replaced with <<junction points>>.

This part was revised as:

<< From structural point of view, it is important that the intermolecular interactions that ensure the junction points recover rapidly and in a high extent after printing. The self-healing ability is given by the presence of dynamic reversible bonds in a network which are destroyed when high forces are applied and then reestablished in time. Such reversible bonds could be non-covalent physical bonds (such as hydrogen bonding, hydrophobic, host-guest, electrostatic interactions, π–π stacking, metal coordination [114,115] or covalent bonds (such as disulfide or imine bonds, boronic esters, reversible diarylbisbenzo-furanone crosslinking, urea/urethane exchange [115–119].>>

  1. In chapter 3.4 the sentence “Tailorable properties can be influenced by multiple crosslinking strategies using combined methods, dynamic or hybrid materials […]” is very unclear. What are crosslinking methods that use dynamic materials? What are dynamic materials? What are hybrid materials? What is a combined method in this context? It is not explained in more detail in the following what these terms should signify.

The sentence was revised.

<< Tailorable properties can be induced by multiple crosslinking strategies using combined methods [2,13,127–133], the incorporation of dynamic reversible bonds [134,135] or the design of hybrid polymer/peptide materials [54,93] that allow different relaxation mechanisms during bioprinting and ensure the shape fidelity. Recommended recovery times for bioprinting are of the order of tens of seconds (Figures 6 and 7), as for example above 85 % recovery of elastic modulus or viscosity within up to 30 s [77,136]. >>

As it was mentioned above, the sentence was improved:

<< From structural point of view, it is important that the intermolecular interactions that ensure the junction points recover rapidly and in a high extent after printing. The self-healing ability is given by the presence of dynamic reversible bonds in a network which are destroyed when high forces are applied and then reestablished in time. Such reversible bonds could be non-covalent physical bonds (such as hydrogen bonding, hydrophobic, host-guest, electrostatic interactions, π–π stacking, metal coordination [114,115] or covalent bonds (such as disulfide or imine bonds, boronic esters, reversible diarylbisbenzo-furanone crosslinking, urea/urethane exchange [115–119]. >>

The indicated references discuss these aspects, here they were very briefly mentioned.

The rest of the critical points are just referring to grammar, spelling and English which should be checked once more. Some words are just misspelled, which happens easily, I will just give 5 examples, but the whole text should be read carefully once more:

  1. Line 218, “dissiped” should be dissipated, Line 232, “bellow” should be below, Line 234 “facilitate” should carry the s to facilitates, Line 237 “monay” should be money, Line 362 “for a various materials” should drop the a and read for various materials.

If these points are addressed and the review is rewritten, it is an excellent starting point for people dealing with biofabrication, especially when they are using extrusion-based techniques. It was fun reading some of the parts and just a little more clarity will make this article very recommendable to a broad audience. I suggest to accept the article after major revisions.

I am grateful to the reviewer for the deep analysis of the manuscript and constructive comments and suggestions that were helpful to improve the manuscript.

Reviewer 2 Report

The work is very well structured, it is clearly presented and I think it will be useful to specialists in the field. 

The conclusions, correct and original, come to complete a very well-documented work.

Author Response

The work is very well structured, it is clearly presented and I think it will be useful to specialists in the field. 

The conclusions, correct and original, come to complete a very well-documented work.

I thank the reviewer for his appreciation and for his availability to analyze the manuscript.

Reviewer 3 Report

The review by Bercea M. focuses on the rheological properties of soft materials that can be used in additive technologies. The author provides data on existing additive manufacturing methods, describes possible rheological characteristics, and provides information on existing literature on the use of soft materials in additive applications. It should be noted that the review is quite long since it contains a lot of basic rheological information. Before being published, the review should be corrected according to the following comments.

Line 83: “Low viscous materials lead to soft and less stable shapes.” Viscosity has nothing in common with the "softness" of the material and its form stability, which are determined by the elastic characteristics of the material: its storage modulus (stiffness) and loss tangent (elasticity).

Line 135: There is a typo: “vaxes” -> “waxes”.

Line 174: “milimeters” –> “millimeters”.

Line 237: “monay” -> “money”

Line 249-254 or 264-271: It should be noted here that the highest Newtonian viscosity may not exist, i.e., that there are materials that do not flow at shear stresses less than the yield stress.

Line 289: “A higher viscosity of the ink improves its mechanical properties”. This is not an obvious statement. The author should explain the mechanism by which an increase in viscosity causes an increase in mechanical properties.

Line 307: “the macromolecules can stick together or on the extruder walls”. Macromolecules cannot stick to each other. At high shear rates, macromolecules can stop moving with respect to each other due to the tightening in macromolecular entanglements (like a filament can stop pulling out of a non-dense knot at high speed, see 10.1007/s00397-011-0556-z.).

Line 324: “solid like-properties” -> “solid-like properties”.

Lines 334-340, 345-352. Here a simple case is considered when a material has only a single yield stress, whereas there can be two of them in the general case: dynamic and static yield stresses, where one corresponds to the destruction of the material structure, and the second to the maintenance of the structure in the destroyed form (see 10.1039/B517840A, 10.1016/j.triboint.2020.106318, 10.1016/j.cemconcomp.2017.11.019, 10.1016/j.molliq.2022.119872). It should also be noted.

Line 360: “mentain” -> “maintain”.

Line 404: “overall strength” - > “overall stiffness”.

Line 413: “viscous (G”) are usually” -> “viscous (G”) moduli are usually”.

Line 424: “for networks” -> “for gel networks”.

Line 432: “diferentiation” -> “differentiation”.

Line 442: “Very recently it was shown that physical networks with strong intermolecular interactions behave as elastic solids for shear stress values below the yield stress and as viscoelastic fluids above σo [53].” That has been previously reported in other works (e.g. 10.1016/j.carbpol.2021.118509), which should also be reflected, as long as this statement is necessary.

Line 464: “tixotropy” -> “thixotropy”.

Line 488: “The thitotropy test can be corrobored” -> “The thixotropy test can be corroborated”.

Line 526: “The most important parameters for the homogeneity and stability of ink droplets is the elastic modulus (G’) at high frequency (around 5 kHz [169].” An additional clarification is needed here about the importance of high frequency.

Line 530: “are is” –> “are”.

Line 533: “ow concentrations” -> “Low concentrations”.

Line 543: “G:” -> “G"”.

Line 547: “At high shear rates, nanocellulose chains in suspension align and form shear thinning/thickening, thixotropic systems with rapid recovery [178].” Although the description is correct, the work cited does not consider nanocellulose (shear thinning and shear thickening behaviors of nanocellulose dispersions can be found in 10.1016/j.carbpol.2021.118660, 10.1016/j.triboint.2022.108080).

Author Response

The review by Bercea M. focuses on the rheological properties of soft materials that can be used in additive technologies. The author provides data on existing additive manufacturing methods, describes possible rheological characteristics, and provides information on existing literature on the use of soft materials in additive applications. It should be noted that the review is quite long since it contains a lot of basic rheological information. Before being published, the review should be corrected according to the following comments.

I thank the reviewer for his appreciation and for his availability to analyze the paper.

Line 83: “Low viscous materials lead to soft and less stable shapes.” Viscosity has nothing in common with the "softness" of the material and its form stability, which are determined by the elastic characteristics of the material: its storage modulus (stiffness) and loss tangent (elasticity).

I thank the reviewer for this comment. The sentence was corrected.

<< The applied shear stress and shear thinning behavior under well-established conditions of shear rates must be examined for determining the printability of bioinks. Low viscous materials lead to soft and less stable shapes. In addition, the viscoelastic parameters (G’, G” and tand) provide information on structure and stability of inks before and after printing process. >>

Line 135: There is a typo: “vaxes” -> “waxes”.

This sentence was removed.

Line 174: “milimeters” –> “millimeters”.

The sentence containing this mistake was removed.

Line 237: “monay” -> “money”

The correction was made.

Line 249-254 or 264-271: It should be noted here that the highest Newtonian viscosity may not exist, i.e., that there are materials that do not flow at shear stresses less than the yield stress.

I thank the reviewer for this comment. The following comment was added:

<< There are materials for which ho is not experimentally accessible; these systems flow at very low shear stress values.>>

Line 289: “A higher viscosity of the ink improves its mechanical properties”. This is not an obvious statement. The author should explain the mechanism by which an increase in viscosity causes an increase in mechanical properties.

Thanks to reviewer, the correction was made: << A higher viscosity and network strength improves the mechanical properties of the ink. >>

Line 307: “the macromolecules can stick together or on the extruder walls”. Macromolecules cannot stick to each other. At high shear rates, macromolecules can stop moving with respect to each other due to the tightening in macromolecular entanglements (like a filament can stop pulling out of a non-dense knot at high speed, see 10.1007/s00397-011-0556-z.).

The indicated reference has been consulted and a correction was made:

<< High values of k and h are associated with hard extrusion of materials from the nozzle; at high shear rates, inhomogeneous flows appear [84]...>>

Line 324: “solid like-properties” -> “solid-like properties”.

The correction was made in the revised manuscript.

Lines 334-340, 345-352. Here a simple case is considered when a material has only a single yield stress, whereas there can be two of them in the general case: dynamic and static yield stresses, where one corresponds to the destruction of the material structure, and the second to the maintenance of the structure in the destroyed form (see 10.1039/B517840A, 10.1016/j.triboint.2020.106318, 10.1016/j.cemconcomp.2017.11.019, 10.1016/j.molliq.2022.119872). It should also be noted.

A new sentence was introduced and new references were added:

<< Some papers reported the existance of two values of so: a dynamic yield stress corresponding to the destruction of the material structure and a static yield stress that maintain the structure in the disturbed state [89–92]. Better results for yield stress are obtained by presetting the shear force in controlled shear stress tests and they are influenced by the shear history and the used evaluation methods [88]. >>

The following corrections were made in the revised manuscript, I thank the reviewer:

Line 360: “mentain” -> “maintain”.

Line 404: “overall strength” - > “overall stiffness”.

Line 413: “viscous (G”) are usually” -> “viscous (G”) moduli are usually”.

Line 424: “for networks” -> “for gel networks”.

Line 432: “diferentiation” -> “differentiation”.

Line 464: “tixotropy” -> “thixotropy”.

Line 488: “The thitotropy test can be corrobored” -> “The thixotropy test can be corroborated”.

Line 530: “are is” –> “are”.

Line 533: “ow concentrations” -> “Low concentrations”.

Line 543: “G:” -> “G"”.

Line 442: “Very recently it was shown that physical networks with strong intermolecular interactions behave as elastic solids for shear stress values below the yield stress and as viscoelastic fluids above σo [53].” That has been previously reported in other works (e.g. 10.1016/j.carbpol.2021.118509), which should also be reflected, as long as this statement is necessary.

This new reference was added, I thank the reviewer for the recommendation:

Line 526: “The most important parameters for the homogeneity and stability of ink droplets is the elastic modulus (G’) at high frequency (around 5 kHz [169].” An additional clarification is needed here about the importance of high frequency.

Thanks to the referent for the comment. The sentence was revised.

<< Determination of the shear viscosity is not enough for the jetting performances, short timescales parameters being also necessary to evaluate the stability of ink droplets. Thus, the jetting behavior was correlated with the elastic modulus determined at 5 kHz (using a special device, a piezoelectric axial vibrator rheometer) [188]. >>

Line 547: “At high shear rates, nanocellulose chains in suspension align and form shear thinning/thickening, thixotropic systems with rapid recovery [178].” Although the description is correct, the work cited does not consider nanocellulose (shear thinning and shear thickening behaviors of nanocellulose dispersions can be found in 10.1016/j.carbpol.2021.118660, 10.1016/j.triboint.2022.108080).

The two references were added, I thank the reviewer for the recommendation.

I am grateful to the reviewer for the deep analysis of the manuscript and constructive comments that were helpful to improve the manuscript.

Round 2

Reviewer 1 Report

Many thanks to the author for reworking the whole manuscript in this swift and comprehensible way. Content-wise and structurally I have no further points that need to be improved.

Lastly, the author should once check the quality (resolution) of all Figures. In the revised manuscript they seem to be inserted at a much lower resolution than the original manuscript's Figures.

Other than that, I would suggest the manuscript for publication.

Reviewer 3 Report

The author has made the necessary corrections to the manuscript with care. In my opinion, it can now be published in its current form.